# Myopericarditis in a Male Adolescent Following the Second Shot of COVID-19 Pfizer m-RNA Vaccine: Typical Example and Analysis of 110 Single Case Reports

Alessandra Piras [1], Giada Melis [2], Lucia Cugusi [3] and Pier Paolo Bassareo [4,5,6,*]

1 Struttura Complessa di Cardiologia-UTIC-Emodinamica, Azienda Ospedaliero-Universitaria di Cagliari, 09042 Monserrato, Cagliari, Italy
2 Pronto Soccorso ed OBI, Azienda Ospedaliero-Universitaria di Cagliari, 09042 Monserrato, Cagliari, Italy
3 Department of Biomedical Sciences, University of Sassari, 07100 Sassari, Sardinia, Italy
4 School of Medicine, University College of Dublin, D04 V1W8 Dublin, Ireland
5 Mater Misericordiae University Hospital, D07 R2WY Dublin, Ireland
6 Children's Health Ireland at Crumlin, D12 N512 Dublin, Ireland
* Correspondence: piercard@inwind.it

**Abstract:** One of the most powerful weapons against COVID-19 is vaccines. After the worldwide spread of the disease, m-RNA vaccines were authorized not only in adult patients, but also in children and adolescents aged 12–18. Since then, alarming reports of cases of myocarditis and/or pericarditis have been noted, primarily involving males after the second vaccine shot. A typical example of myopericarditis occurring in an adolescent a few days after the second shot of an m-RNA vaccine is described here. An in-depth review of all 110 single case reports published up to July 2022 with related features and outcomes is also presented. This is the first extensive analysis focused solely on a significant number of single case reports, which have usually been excluded from systematic reviews and meta-analyses carried out in the field. The analysis presented here confirms that most cases occurred in males after the second injection of an m-RNA vaccine. Cases were mild and responsive to the usual medical treatment. What is newly reported is that not only adolescents, but also older people, especially females, were affected by this adverse event.

**Keywords:** m-RNA vaccine; COVID-19; myocarditis; pericarditis; adverse event

## 1. Introduction

Since the end of 2019, the world has been coping with the widespread coronavirus disease (COVID-19) pandemic and its harmful consequences [1]. With the aim of reducing fluctuant outbreaks of COVID-19, European and American authorities have decided to extend vaccination to adolescents [2]. Adolescents are potential carriers of coronavirus, and may even suffer from a multi-inflammatory syndrome (MIS-C) triggered by the virus [3]. However, many reports have highlighted the risk of the occurrence of myocarditis and/or pericarditis after a COVID-19 m-RNA vaccine (BNT162b2 mRNA by Pfizer-BioNTech and m-RNA 1273 by Moderna) injection, although a direct link between the two is still a subject of debate [4], and investigations are ongoing. The risk is primarily for adolescent male subjects aged 16 or over a few days following their second vaccine dose [5]. Although those receiving an m-RNA vaccine (vs. a non m-RNA vaccine) are at higher risk of developing myo/pericarditis, those receiving adenovirus vector vaccines, such as AstraZeneca or Janssen/Johnson & Johnson, are not without risk [6].

The first warning was released by the US Center for Disease Control and Prevention (CDC) in May 2021 [7]. The EMA's Pharmacovigilance Risk Assessment Committee (PRAC) immediately noted that they were investigating reports of myo/pericarditis after the Pfizer-BioNTech vaccine, but did not notice any concerns, as the detected rate of such side effects was similar to that in the general population [4]. On June 2021, Israeli health

regulators reported 275 events of myocarditis between December 2020 and May 2021 among vaccinated people, mostly males aged 16–24, who had received the Pfizer COVID-19 vaccine [8,9]. Since then, many case reports and a few retrospective case series in the field were reported in the literature [10]. The first prospective cohort study was by Mansanguan et al., who found seven cases of proven myocarditis among 301 children and adolescents aged 13–18 following their second inoculation with Pfizer-BioNTech vaccine [11].

Case series have generally been used for systematic reviews and meta-analyses published in the field, whereas single case reports have been excluded from analysis.

The aims of this paper are to present the paradigmatic case of a young male adolescent who developed myopericarditis approximately two weeks after receiving a second dose of Pfizer vaccine and reviewing the literature to date regarding single case reports of myocarditis and/or pericarditis triggered by COVID-19 vaccination, with related features and outcomes.

## 2. Case Report

A 16-year-old male presented at the Admission and Emergency Department owing to the onset of a progressively increasing heavy/tight chest pain. He reported receiving a second dose of COVID-19 Pfizer vaccine 13 days previous. The patient developed fever, diffuse arthromyalgia, and asthenia and was admitted to hospital. On cardiovascular examination, tachycardic heart sounds 1 and 2 were audible, along with a ubiquitous grade 1/6 intensity ejection systolic murmur. The patient's chest was clear, with good air entry bilaterally. Ankle oedema was ruled out. Femoral pulses were palpable bilaterally. Blood pressure values were low, at 97/58 mmHg. Laboratory testing showed an increase in white blood cells, raised ESR and CRP, and significantly elevated troponin (12,000 ng/L, range: 0.01–35 ng/L). Immunoglobulin G against COVID-19 was increased, and a nasopharyngeal swab was negative. The patient's ECG displayed diffuse ST elevation (see Figure 1).

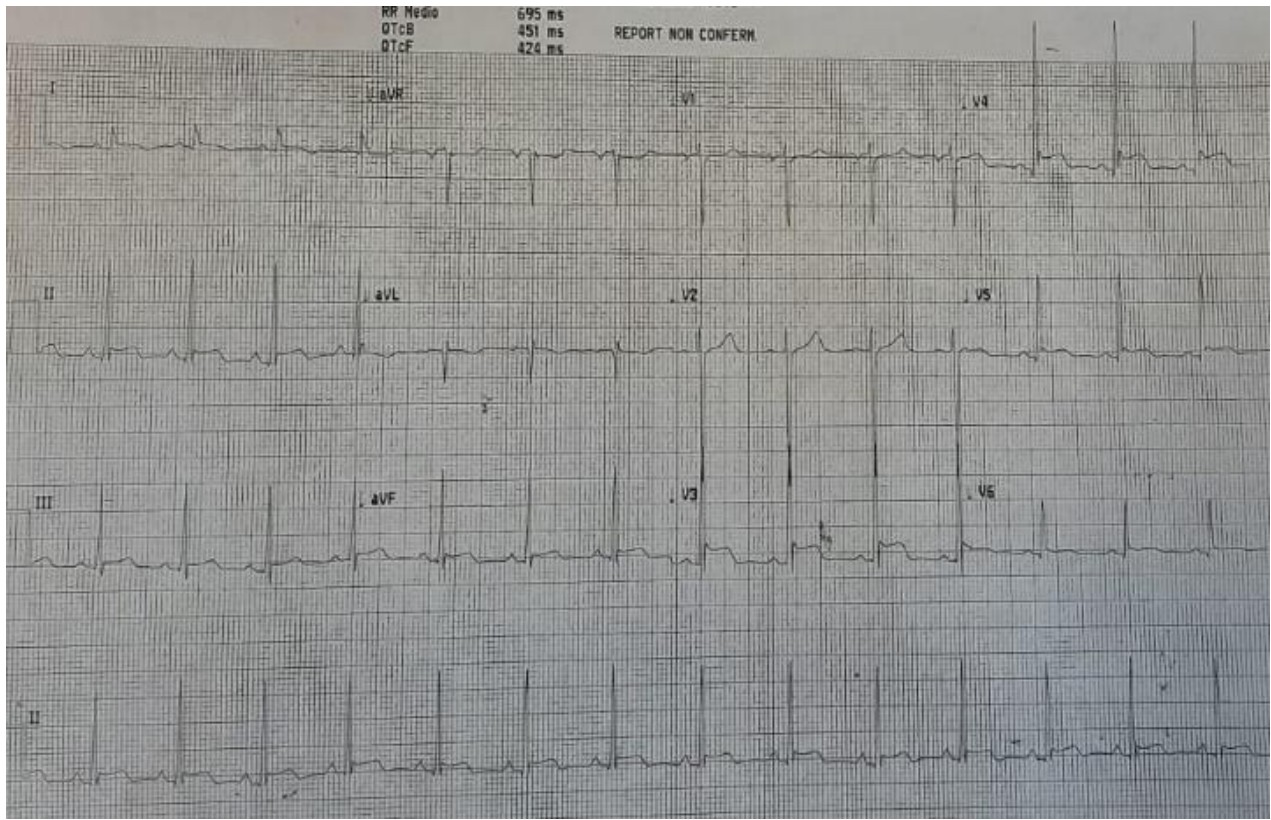

**Figure 1.** ECG at admission, with diffuse ST elevation.

At echocardiography, the left ventricle size was normal. However, a severe systolic dysfunction (ejection fraction 32%) was noted, owing to akinesia of the inferior wall (basal and mid segments), posterior wall, septum, and anterior wall (mid and apical segments). There was a mild mitral valve regurgitation as well as a mild circumferential pericardial effusion. Acute coronary syndrome was excluded due to the patient's young age, lack of risk factors, recent vaccination, positivity to anti COVID-19 IgG, and the onset of high temperature. Owing to the suspicion of myocarditis, treatment with IV lysine acetylsalicylate 1 g three times/day, oral colchicine 0.5 mg twice/day (it was stopped after a week, due to a harmful increase in transaminases), i.v. ceftriaxone 1 g/day, i.v. furosemide 20 mg/day (for 2 days) was started. A proton pump inhibitor (esomeprazole 40 mg/day) was also added. On the second day of hospitalization, a cardiac magnetic resonance (CMR) with late gadolinium enhancement (LGE) was performed, which displayed the occurrence of myopericarditis (Figure 2). A few studies have proven that an LGE > 15% of the total myocardial mass is linked to an increased risk of sudden death in other cardiomyopathies, such as hypertrophic cardiomyopathy [12].

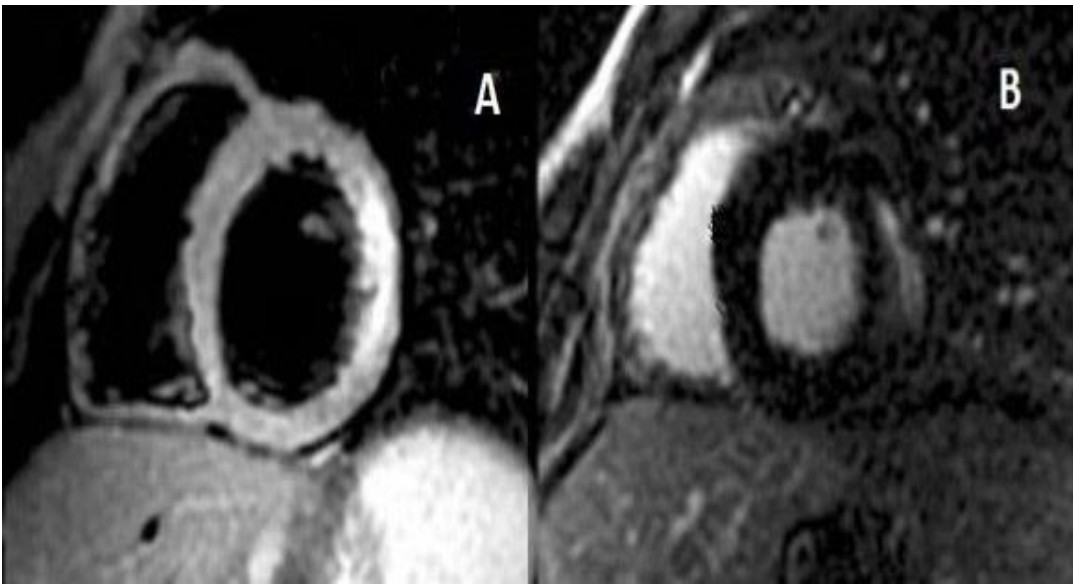

**Figure 2.** Panel (**A**) Short axis T2-STIR image with lateral hyperintense signal due to myocardial oedema. Panel (**B**) Lateral subepicardial area of late gadolinium enhancement (18% of the left ventricle).

A week after treatment started, the chest pain resolved and the patient's energy levels gradually normalized. An ultrasound scan revealed that left ventricular function and related wall motion abnormalities were fully recovered and that pericardial effusion had disappeared. In addition, troponin normalized and inflammatory markers significantly improved, although they remained slightly elevated. The patient did not require admission to the intensive care unit. He was discharged 10 days after admission; follow-up and rehabilitation were planned. Figure 3 presents the ECG at discharge.

As per Italian law, side effects of the Pfizer vaccine were reported to the Agenzia Italiana del Farmaco (AIFA).

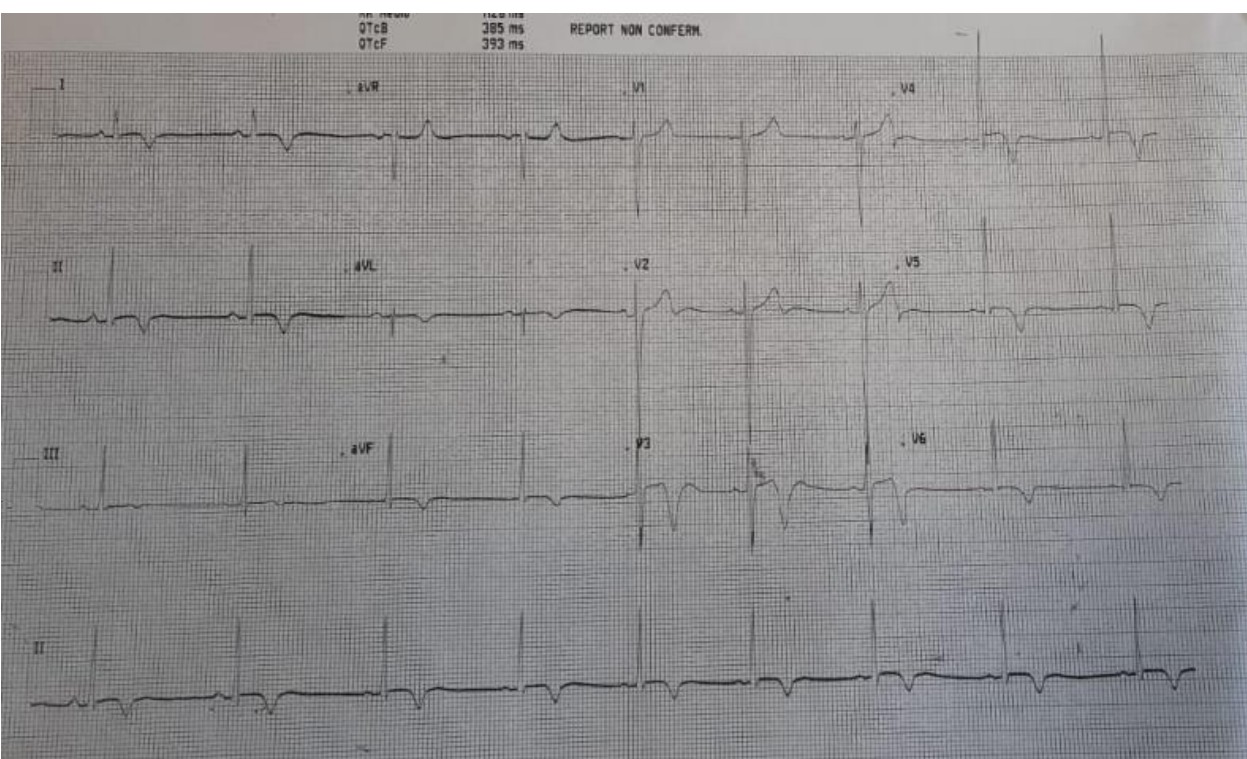

**Figure 3.** ECG at discharge. Diffuse ST elevation resolved and was replaced by deep negative T waves.

### 3. Search Strategy

The electronic databases PubMed, EMBASE, and Scopus were searched from their establishment up to 22 July 2021. The MeSH (Medical Subject Headings) search terms were "case report" and/or "myocarditis" and/or "myocardium" and/or "pericarditis" and/or "pericardium" and/or "myopericarditis" and/or "acute myocardial injury" and/or "troponin" and/or "acute coronary syndrome" and/or "COVID vaccine". Unlike previous meta-analyses in the literature, which excluded single case reports [13], only single case reports have been included in this analysis. On the other hand, we excluded case series, meta-analyses, animal studies, and papers written in languages other than English.

#### 3.1. Study Selection

Two authors (A.P. and G.M.) separately reviewed the selected abstracts and evaluated whether they were eligible. Full-texts were checked when both reviewers thought the article might match the inclusion criteria. Any possible dispute was resolved with the involvement of a third author (L.C.).

#### 3.2. Data Extraction

Three authors (A.P., G.M., and L.C.) independently extracted information from selected single case reports, including age, sex, vaccine type, vaccine dosage, length of time between vaccination and disease occurrence, symptoms, ECG on admission, troponin, imaging (echocardiogram and cardiac MRI), biopsy, and outcome.

#### 3.3. Data Presentation

Data were presented in the form of mean ± SD. The chi–square test and Mann–Whitney U test were used to check statistical significance when needed. Statistical significance was set to $p < 0.05$.

## 4. Results

Overall, 229 potential single case reports of myocarditis and/or pericarditis were identified using the PubMed, Embase, and Scopus databases. Forty were duplicates. Another 79 papers were excluded after checking their abstracts. The remaining 110 case reports were analysed regarding patients' characteristics and disease outcomes. These reports included 12 cases of pericarditis (10.9%), 20 cases of peri-myocarditis/myo-pericarditis (18.2%), and 78 cases of myocarditis (70.9%). The latter were prevalent compared with pericarditis and peri-myocarditis combined ($p < 0.00001$) (Figure 4).

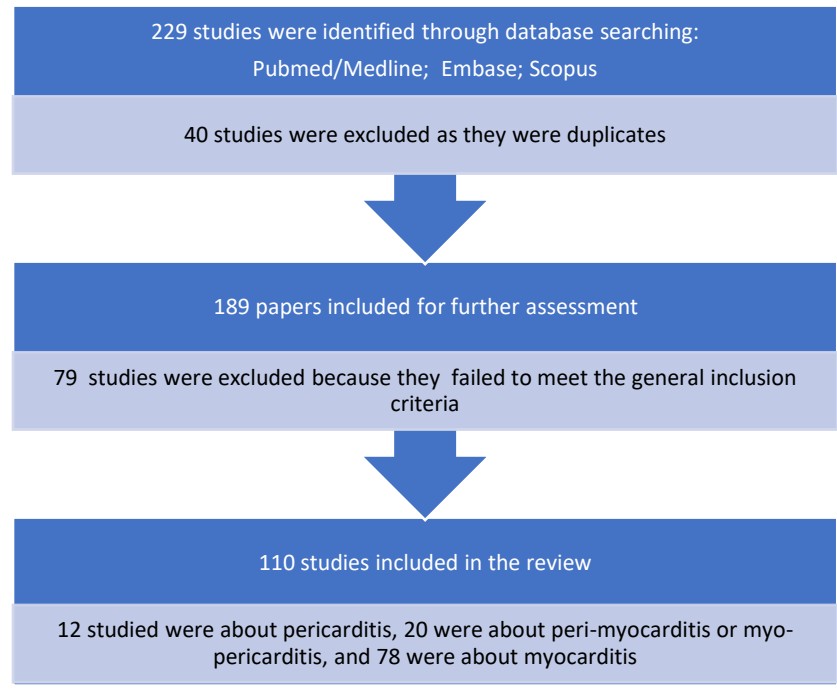

**Figure 4.** Case reports selection process.

The vast majority of cases reported occurred in males (75.4%; male-to-female ratio 3.1:1) under the age of 40 (36.5 ± 18.1 years), after the second dose (64.5%) of Pfizer-BioNTech vaccine (50.5%). Another m-RNA vaccine, e.g., Moderna, was responsible for 33.6% of cases. Very few cases were linked with taking an AstraZeneca or Janssen/Johnson & Johnson vaccine. For male patients the mean age at onset was 31.7 ± 15.5 years, whereas female patients were older (51.6 ± 17.4 years). This was statistically significant ($p < 0.00001$). Notably, pericarditis cases were balanced between genders (7 males vs. 5 females, $p$ = ns). The onset of symptoms occurred after 5.7 ± 9.6 days after the vaccine shot. The most usual symptom was chest pain (67.3%). The most commonly detected ECG change was ST tract elevation (51.8%). Troponin was tested in 91.8% of cases and found to be increased in 90.1% of cases. Echocardiography was performed in 93.6% of cases and cardiac MRI was performed in 69.1% of cases. Echocardiograms showed that ejection fraction was reduced in 38.8% of patients. Cardiac biopsy was performed in a minority of patients with myocarditis (34.6%). The outcome was favorable in most cases (62.9%); death or required heart transplantation was the outcome in 5.4% of cases.

Most cases were attributable to the Pfizer-BioNTech vaccine (See Table 1).

**Table 1.** Type of vaccine and occurrence of myocarditis and/or pericarditis in the reported case series.

| Pfizer-BioNTech | Moderna | AstraZeneca | Other/and Janssen Not Specified |
|---|---|---|---|
| Number: 60/110 | 37/110 | 8/110 | 5/110 |
| Percentage: 54.5% | 33.7% | 7.3% | 4.5% |

See supplementary material for a list of the 110 case reports analysed and related references (see Table S1).

## 5. Discussion

The 29 June 2021 issue of *JAMA Cardiology* included four papers (two reports and two editorials) regarding the onset of myocarditis in teenagers following the second injection of a COVID-19 m-RNA vaccine. This was the first advice concerning the occurrence of a potential vaccine-triggered adverse event [14–17]. Since then, case series and numerous case reports about the same topic have been published [18].

Findings of the present analysis confirm what is already widely known: there is an association between the second shot of an m-RNA vaccine and the possible onset of myocarditis, primarily in males. The outcome is for the most part favorable, although a few heart transplantations or deaths were reported. What is new, compared to reports noted in the literature to date, is that patients in older age groups are also at risk of this adverse event. This is particularly evident regarding middle-aged women. Troponin was significantly increased in most patients. Cardiac MRI reports were consistent with the identification of myocarditis in more than two thirds of cases.

Myocarditis as a vaccine side effect was anecdotal before COVID vaccination, and scientifically proven only for vaccines against smallpox, which differ significantly from m-RNA vaccines against COVID-19 [17].

The short time lapse between m-RNA vaccination and myocarditis suggests an association between the two; however, the link is not yet scientifically proven. Further investigation is needed [19], as a direct link between m-RNA vaccination and myocarditis is far from being fully elucidated. Several mechanisms have been suggested, but none is convincing on its own. The many suggested mechanisms include: hyper immune or inflammatory reaction after exposure to spike protein, mRNA strand, or unexplained trigger; late onset of hypersensitivity (serum sickness); eosinophilic myocarditis; hyperreactivity to vaccine excipients (e.g., polyethylene glycol and tromethamine or lipid nanoparticle sheath); reaction to mRNA vaccine lipid nanomolecules; self-immunity through mimicry or other pathways; low residual quantity of double strand RNA; unbalanced micro-RNA reaction; release of anti-idiotype antibodies against certain regions of antigen-specific antibodies; activation of pre-existent dysregulated immune pathways in predisposed subjects (leading to polyclonal B cell growth, immune complex development, and inflammation); antibody-dependent amplification of immunity or other forms of immune intensification with re-exposure to the virus after vaccine inoculation; direct cell invasion through the interplay of the spike protein and the angiotensin converting enzyme 2 (ACE2) widely expressed on cardiomyocytes surfaces; cardiac pericyte expression of ACE2 with disabled immune complex on the surface of pericytes along with activation of the complement system; hyperviscosity-induced cardiac problem; and demanding effort induced secretion of proinflammatory interleukin 6 [13–39]. A combination of some of these immune-mediated mechanisms is likely involved in m-RNA vaccine-induced myocarditis and pericarditis, of which the first manifestation can be fatal.

Notably, the prompt detection of this rare side effect indicates that the surveillance system established after vaccines were released into the market is very efficient [17]. This is important, as a very recent paper proved a prevalence of myocarditis of 537.1/1,000,000 in men aged 18–24 years, which is significantly higher than that reported to US advisory committees [40].

COVID-19 vaccination, especially in pediatrics, certainly needs further investigations regarding its advantages and disadvantages in terms of safety, as its scientifically proven benefits are less evident than in adulthood. This is particularly true as myocarditis, although mild, may lead to the development of intramyocardial fibrosis and scarring, which are potential sites for the onset of life-threatening ventricular arrhythmias later in life [41]. Therefore, on-going surveillance is required, even in cases of apparent full recovery. This need for surveillance is supported by alarming reports of an increased number of deaths

in male competitive athletes after receiving a COVID-19 vaccine; these deaths seems to be related to a combination of subclinical myocarditis induced by the vaccine and a hyper-catecholaminergic state leading to life-threatening ventricular arrhythmias during sporting activities [42].

Therefore, a public health campaign informing high-risk groups of the risk of fatal myocarditis is warranted, and vaccination consent forms should be updated to include fatal myocarditis among possible adverse events.

**Supplementary Materials:** The following supporting information can be downloaded at: https://www.mdpi.com/article/10.3390/pediatric14040048/s1, Table S1: List of the 110 case reports which have been analysed and extracted information.

**Author Contributions:** Conceptualization, A.P. and G.M.; writing—original draft preparation, A.P. and G.M.; writing—review and editing, L.C. and P.P.B.; supervision, P.P.B. All authors have read and agreed to the published version of the manuscript.

**Funding:** This research received no external funding.

**Institutional Review Board Statement:** Not applicable.

**Informed Consent Statement:** Acquired. It is with the first author. The study is based on 110 already published case reports. A new case report has been described as an example for the Readers. The consent to publish such a case is with the first author.

**Data Availability Statement:** Not applicable.

**Acknowledgments:** We thank the patient's parents for providing consent to publish this case report.

**Conflicts of Interest:** The authors declare no conflict of interest.

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
