# Peer review of "Myopericarditis in a Male Adolescent Following the Second Shot of COVID-19 Pfizer m-RNA Vaccine: Typical Example and Analysis of 110 Single Case Reports"

_pediatrrep, doi:10.3390/pediatric14040048_

Round 1
Reviewer 1 Report
Myopericarditis in a male adolescent after the second shot of 2
COVID-19 Pfizer m-RNA vaccine: typical example and analysis 3
of 110 single case reports by Piras et. al. looks well written review.
But I have few comments:
1- figures 1, 2 and 3 should be in better resolution (not as seen like screenshot).- its used in very low low resolution and can't really be understandable
2- The authors need to add a table for showing the percentage of the incidence of the Myopericarditis/myocarditis/pericarditis especially if we have different types of vaccines (moderna/pfizer/Astrazeneca/ Janssen) will be more powerful and attractive.
I think that if nr-2 will not be added it looks more like a "case report" and literature review instead of a real "review report"
Author Response
Dear Colleague,
Thanks for the time spent in reviewing our paper. Your suggestions have been addressed as follows:
- The resolution of the figures is 300 dpi as requested by the Journal. According to your suggestion the images have been magnified;
-
A table (Table 1) highlighting the percentage of peri/myocarditis attributable to each kind of vaccine has been added as per your advice (Results).
The revised text is in red. Other changes have been made according to the other Reviewers' requests.
Reviewer 2 Report
A well-written case report, although there was no direct evidence of mRNA vaccine results in myopericarditis, authors tried their best to review the existing publications.
Author Response
Dear Colleague,
Thanks for the time spent in reviewing our paper and your appreciation.
The fact that a direct link between getting Covid m-RNA vaccine and the onset of peri/myocarditis is still far from being 100% sure has been already highlighted in the Introduction (page 2, line 39) and Discussion (page 7, lines 201-203) sections.
Some changes to the text have been made according to the other Reviewers' suggestions. They are in red.
Reviewer 3 Report
Case Description: Please give the percent of the LV detected as "scar" by MRI. The figure depicts a large zone >15%. Please discuss the importance of large zones of LGI and risk of sudden death in other cardiomyopathic states.
Introduction. Please mention and cite the data from Sharff et al, who found a peak rate of myocarditis in men ages 18-24 at 537.1/million.
Introduction. Please mention and cite the first prospective cohort study by Mansanguan who found 7/301 of children ages 13-18 met a definition of myocarditis after the second injection of Pfizer/BNT
Introduction. Please correct statements to indicate the adenoviral vaccines have been associated with myocarditis. Cite Ling et al Lancet Resp Med
Discussion. Please mention and cite the published FATAL cases of myocarditis by Choi, Verma, and Gill and indicate that the initial presentation with a few days of taking the COVID-19 vaccine can be death.
Discussion. Please mention the concern over athletes developing subclinical myocarditis after COVID-19 vaccination and then the risk for catecholamine induced sudden death with sports. Cite Cadegiani et al, preprint 2022.
Discussion. Suggest that there should be a public health campaign warning high risk groups of fatal myocarditis. Also suggest the consent forms be updated to include fatal myocarditis.
Author Response
Dear Colleague,
Thank for the time spent in reviewing our paper. All your suggestions have been addressed carefully, namely:
- The percentage of scar in the patient’s LV has been reported now (Figure 2) according to your suggestion. A comment with reference concerning the link between late gadolinium enhancement over 15% in other cardiomyopathies like HCM and risk of sudden death has been added as well [ref 23];
-
The paper by Sharff and Coll. has been cited (ref 40, Discussion);
-
The paper by Mansanguan and Coll. has been cited (ref 11, Introduction);
-
The paper by Ling and Coll. has been cited (ref. 6, Introduction);
-
The cases by Choi, Verma, and Gill have been already cited in the first version of the paper [see references 129 in Supplementary material, 38 in the main test, 56 in Supplementary material, respectively]. The statement that the initial presentation may be death, although already outlined, has been reinforced (Discussion, page 8, line 223);
-
The paper by Cadegiani has been cited (ref 42, Discussion);
-
The suggested important statements have been added (Discussion, at the very end).The revised text is in red. Other changes have been made according to the other Reviewers' advice.